# Thrombectomy-Capable Stroke Centre—A Key to Acute Stroke Care System Improvement? Retrospective Analysis of Safety and Efficacy of Endovascular Treatment in Cardiac Cathlab

**DOI:** 10.3390/ijerph20032232

**Published:** 2023-01-26

**Authors:** Krzysztof Pawłowski, Artur Dziadkiewicz, Anna Podlasek, Jacek Klaudel, Alicja Mączkowiak, Marek Szołkiewicz

**Affiliations:** 1Department of Cardiology and Interventional Angiology, Kashubian Center for Heart and Vascular Diseases, Pomeranian Hospitals, 84-200 Wejherowo, Poland; 2Department of Neurology and Stroke, Pomeranian Hospitals, 84-200 Wejherowo, Poland; 3Tayside Innovation Medtech Ecosystem (TIME), University of Dundee, Dundee DD1 4HN, UK; 4Precision Imaging Beacon, Radiological Sciences, University of Nottingham, Nottingham NG7 2RD, UK; 5Department of Invasive Cardiology, St. Adalbert’s Hospital, Copernicus PL, 80-070 Gdansk, Poland

**Keywords:** acute ischaemic stroke, mechanical thrombectomy, thrombectomy-capable stroke centre, endovascular stroke treatment

## Abstract

The optimal structure of the acute ischaemic stroke treatment network is unknown and eagerly sought. To make it most effective, different treatment and transportation strategies have been developed and investigated worldwide. Since only a fraction of acute stroke patients with large vessel occlusion are treated, a new entity—thrombectomy-capable stroke centre (TCSC)—was introduced to respond to the growing demand for timely endovascular treatment. The purpose of this study was to present the early experience of the first 70 patients treated by mechanical means in a newly developed cardiac Cathlab-based TCSC. The essential safety and efficacy measures were recorded and compared with those reported in the invasive arm of the HERMES meta-analysis—the largest published dataset on the subject. We found no significant differences in terms of clinical and safety outcomes, such as early neurological recovery, level of functional independence at 90 days, symptomatic intracranial haemorrhage, parenchymal haematoma type 2, and mortality. These encouraging results obtained in the small endovascular centre may be an argument for the introduction of the TCSC into operating stroke networks to increase patient access to timely treatment and to improve clinical outcomes.

## 1. Introduction

Stroke is one of the leading causes of disability and death throughout the civilised world. Mechanical thrombectomy (MT) has revolutionised the treatment of acute ischaemic stroke patients with large vessel occlusion (LVO) [1,2,3,4,5]. It is estimated that 10%–12% of ischaemic stroke patients may be candidates for interventional therapy, which starkly contrasts with the numbers of MT provided with median access of 2.76% worldwide [6,7,8,9].

Changes to stroke network structure and operator training are introduced to meet the growing demand for timely thrombectomy [10,11]. As a result, a new level of stroke care certification in the US was introduced in 2018—a thrombectomy-capable stroke centre (TCSC). Since then, 44 such units have been registered in the US to support 262 comprehensive stroke centres (CSC). Since the introduction of the TCSC, a drop in case fatality rate was observed, while the outcomes achieved were better than that of primary stroke centres (PSC) and comparable to CSC [12,13]. The acute ischaemic stroke system in Poland suffers from small MT penetration (roughly 4% of all ischaemic strokes treated with MT in 2021) and considerable delays due to long distances to CSC [14]. Therefore, we believe there is an urgent need for change in the Polish MT network structure and propose the introduction of thrombectomy-capable stroke centres (TCSC) based on existing upgraded primary stroke centres (PSC). To support the idea, we present the experience of our institution that underwent such a transformation recently [15].

The short- and medium-term results of our first 70 acute LVO ischaemic stroke patients treated by mechanical reperfusion techniques were compared with the data available from the HERMES meta-analysis of five randomised trials (MR CLEAN, ESCAPE, REVASCAT, SWIFT PRIME, and EXTEND IA) [16]. Safety and efficacy results support the idea that experienced and integrated multidisciplinary teams may be a suitable platform to transform PSC into TCSC.

## 2. Materials and Methods

### 2.1. Patients

All patients were scanned with CT/CTA or MRI before the intervention. Our analysis included patients with acute ischaemic strokes due to LVO (plus M2). LVO was defined as the occlusion of the internal carotid artery (ICA), the middle cerebral artery (MCA) at the M1 level, the intracranial vertebral artery (VA), and the basilar artery (BA) in computed tomography angiography (CTA). Other inclusion criteria were as follows: age > 18 years, National Institutes of Health Stroke Scale (NIHSS) ≥ 6 or isolated aphasia, prior functional independence with the modified ranking scale − mRS ≤ 2, time from onset to presentation <6 h. In addition, we also included four patients classified as ‘wake-up’ strokes with time of presentation 6–24 h and a confirmed mismatch between ischaemic core and penumbra on magnetic resonance imaging (MRI). All eligible patients received intravenous thrombolysis (IVT) with recombinant tissue plasminogen activator (rt-PA) prior to intervention. In the case of contraindication to IVT, we proceeded with MT after computed tomography (CT) and CTA. The intervention decision was made by a team consisting of a neurologist, a radiologist, and an interventionalist. We excluded patients with demarcated massive strokes on scanning (more than 1/3 of the territory of the MCA). All patients were Caucasian. Informed consent was taken from all patients before treatment initiation.

### 2.2. MT Technique

A senior interventional cardiologist and a neurologist performed all interventions. The team comprised two interventional cardiologists, an interventional neurologist in training and a vascular surgeon in training. All procedures were performed in an angio suite on a Siemens Artis Zee monoplane system. MT was performed either under conscious sedation (CS) or general anaesthesia (GA). In cooperative patients (except tandem lesions, dissections, and M2 occlusions), conscious sedation was preferred. Balloon guide catheter (BGC) placement in the internal carotid artery (ICA) was our initial strategy; in the cases of difficult access or intervention in the posterior circulation, we used long interventional 6F sheaths. In all the procedures, a distal access catheter (DAC) was introduced intracranially, and in the case of proximal thrombus location, one pass with a direct aspiration first pass technique (ADAPT) technique was attempted. Our default strategy was the stent-retriever assisted vacuum-locked extraction (SAVE) technique, where thrombus extraction was performed using 2-point active suction on both BGC and DAC, combined with a stent retriever [17]. After the procedure, the groin puncture site was rechecked by angiography and, if possible, secured with a vascular plug (or otherwise by manual compression).

### 2.3. Procedure Safety and Efficacy

Angiograms were assessed and classified by 2 interventionalists. We used the modified treatment in cerebral infarction (mTICI) scale to score pre- and post-procedural cerebral flow, with success defined as the TICI 2b/3 score [18]. All patients were scanned by CT or MRI within 24 h post-procedure to assess the extent of the infarcted area and the presence of complications. In all patients who received stents, CT was performed immediately after the procedure. CT was also urgently performed in cases of clinical deterioration. We classified intracranial haemorrhage (ICH) according to the Heidelberg bleeding classification [19]. Symptomatic intracranial haemorrhage (sICH) was defined as ICH with a concomitant drop in NIHSS by at least 4 points. Safety outcomes were classified as the proportion of patients with sICH, parenchymal haematoma type 2 (PH2) within 48 h post-procedure, and 90-day mortality. Efficacy outcomes were determined by estimation of the mean NIHSS score at 24 h, a change in the mean NIHSS score from the baseline to 24 h, and early neurological recovery (ENR) at 24 h (NIHSS drop of ≥8), together with a degree of disability measured by the modified Rankin score at 90 days.

### 2.4. Patient Evaluation

A neurologist assessed patients according to the acute stroke protocol used at our institution. At the baseline—apart from age, sex, and standard clinical data (arterial hypertension, diabetes mellitus, atrial fibrillation, and smoking)—we measured initial NIHSS and the mRS scale. We noted the location of LVO and measured NIHSS at 24 h, as well as mRS at 90 days post-procedure. Crucial patient care times were noted: onset-to-LVO detection on the CTA scan, onset-to-IVT administration, and onset-to-reperfusion on final angiography. Control head CT was routinely performed 24 h post-procedure and urgently in the above-mentioned cases.

### 2.5. Data Analysis

Descriptive data were presented as mean±standard deviation (SD)/number of participants in the group (N) or % (n/N). HERMES data were partially presented as the median (interquartile range, IQR), which was transformed into mean ± SD [20]. The Shapiro–Wilk test was performed to assess data normality, and data were deemed to be normally distributed if *p* > 0.05. To compare two cohorts treated with MT, we performed independent sample *t* tests or U-Mann Whitney tests, as applicable for continuous data, and chi-square tests for categorical data [21].

## 3. Results

We compared the safety and efficacy results of the first 70 acute stroke patients treated with endovascular methods in our institution between August 2020 and September 2022, with 634 patients pooled in the invasive arm of the HERMES meta-analysis. During that period, 1224 patients were admitted to our unit with a stroke diagnosis (see Figure 1), 1059 of which were ischaemic. IVT was given to 271 patients (25%), while 77 patients (7%) were qualified for endovascular treatment (EVT). Note that 7 out of 77 patients were not treated on-site but transferred to the nearest CSC for various technical reasons (patients qualified for MT at the same time, angiosuite unavailable, staffing problems) and thus excluded from the analysis. Six patients had EVT for acute stroke due to a carotid stenosis/occlusion (without intracranial intervention); all of these patients received carotid stents.

We noted that mean age, number of men and women, and baseline mean NIHSS were comparable in the two cohorts. Arterial hypertension and atrial fibrillation were more prevalent in our group compared to HERMES. GA was applied in 51% of patients, and conversion from CS to GA was necessary for 3 of the 70 patients (4%). Groin puncture (our default access) was performed under ultrasound guidance in 76% of patients, and in two instances, brachial access was necessary. Regarding the occlusion site, patients with cervical lesions were much more prevalent in our group than in HERMES (62% vs. 21%, *p* < 0.0001), most likely due to tandem lesions being excluded in the latter cohort. At the same time, M1 occlusion was less frequent in our study compared to HERMES (38.6% vs. 69%, respectively). Distal MCA occlusions (M2) were encountered in similar ratios in both groups (7% vs. 8%, respectively). We treated only one patient with an occlusion site other than anterior circulation, namely, the basilar artery (1%), compared to 2% of such cases observed in HERMES (see Table 1). IVT (rt-PA) was administered in the identical percentage of patients in both groups (83%). We calculated crucial interventional times: onset-to-LVO, onset-to-IVT, and onset-to-reperfusion (five patients, three with ‘wake-up’ strokes and two with missing precise timing data, were excluded from the analysis). The onset-to-LVO detection time (onset to randomisation in the HERMES analysis) was significantly shorter in our group (104 vs. 199 min, *p* < 0.001), while the onset-to-IVT time was longer (120 vs. 103 min, *p* = 0.007), most likely as a result of the fact that the vast majority of patients in our institution were primarily treated in the mothership paradigm (see Table 2 below). The onset-to-reperfusion times were comparable in both cohorts (272 vs. 286 min, *p* = 0.3585, respectively).

By measuring the percentage of patients across mRS groups (0–6) after 90 days, we assessed the efficacy of treatment. We also measured the ratio of patients with excellent clinical outcome (mRS 0–1) at 90 days, good clinical outcome (mRS 0–2) at 90 days, mean NIHSS drop within 24 h, and early neurological recovery at 24 h.

The mean baseline NIHSS, mean NIHSS at 24 h, change in mean NIHSS score from baseline to 24 h were all comparable (see Table 3). Early neurological recovery was nonsignificantly lower in our group (46.2% vs. 50.2%, *p* = 0.6336).

In the presented cohort, we observed similar ratios of patients with excellent clinical outcomes (mRS 0–1) at 90 days (30% vs. 26.9%, *p* = 0.6748, respectively) and nonsignificantly more patients with good clinical outcomes (mRS 0–2) at 90 days (55.7% vs. 46%, *p* = 0.1545, respectively) compared to HERMES (see Table 4).

Regarding the safety outcomes, we compared the rate of symptomatic ICH, parenchymal haematoma type 2, and mortality at 90 days in both groups. We observed a nonsignificantly higher prevalence of sICH in our group compared to HERMES (10% vs. 4.4%, *p* = 0.0801). We noted a similar incidence of PH2 (7.1% vs. 5.1%, *p* = 0.6547) and no difference in mortality at 90 days (17% vs. 15%, 0.8177).

## 4. Discussion

In Poland, a heated interdisciplinary discussion has been taking place concerning the future shape of the acute ischaemic stroke network. As great proponents of the idea of wide MT-network expansion, we aimed to show that the safety and efficacy results achieved in our recently developed, small cardiac Cathlab-based thrombectomy-capable stroke centre were comparable to those reported in the HERMES meta-analysis [15,16]. In this study, we compared the early and late safety and efficacy results of the cohort of the first 70 LVO stroke patients treated by MT in our centre with the pooled data of 634 patients from the invasive arm of the HERMES study. The main findings of this retrospective analysis may be summarised as follows: (1) the safety outcomes measured at 90 days (sICH, PH type 2, and mortality) were comparable in both groups, and (2) the efficacy outcomes, both early (NIHSS at 24 h, NIHSS change at 24 h, and ENR) and late (mRS 0–1 and mRS 0–2 at 90 days), were comparable in both cohorts. In detail, we observed similar ratios of patients with excellent (mRS 0–1) and good (mRS 0–2) clinical outcomes at 90 days.

Compared to HERMES, a nonsignificantly higher prevalence of sICH was noted, and this may be associated with much higher incidence, present in our cohort, of ICA occlusion [22]. A similar incidence of PH2 and no difference in mortality at 90 days were observed in compared groups. A limitation of this study is the lack of access to the HERMES analysis raw data, which prevented us from checking data distribution and using nonparametric tests if required.

Since ‘time is brain’, timely reperfusion has become the Holy Grail of modern acute ischaemic stroke care. In patients with severe strokes due to LVO occlusion, MT has revolutionised the treatment and, with time, has become faster, safer, and more widely accessible. Today, 5%–10% of acute ischaemic stroke patients are treated invasively, but treatment availability differs globally [6,7,8,9]. The main issue to be solved is creating effective acute stroke networks, enabling fast selection of LVO patients, and admitting them to thrombectomy centres in time. Considerable delays are the Achilles heel of many healthcare systems, despite the introduction of various transportation models [23,24,25]. These are applied and further researched according to the local environment in terms of stroke network characteristics and its logistics associated with specific geography [26,27]. There are various transportation strategies: drip-and-ship (DS), mothership (MS), and drive-a-doctor (DD). The DS provides fairly fast access to diagnostics and IVT but at the cost of considerable delays in thrombectomy. The MS improves timely access to mechanical intervention, but it is unavailable for many patients who live in rural areas. The DD, while feasible, is probably the least practical solution due to the questionable availability of ‘on-call’ specialists and the suboptimal procedure environment of an unfamiliar team and angio suite [28,29,30]. Acute ischaemic stroke treatment is highly time-sensitive, and there is evidence that cutting down the time to reperfusion improves the outcomes in patients with LVO [31,32]. Moreover, in many cases, direct transport to locally available thrombectomy facilities makes it possible to invasively treat patients with LVO; otherwise, they would be denied treatment due to lengthy transportation times in the DS model [7]. In stroke communities throughout the world, there is an ongoing discussion on how to improve results in the treatment of ischaemic stroke patients. The key question is how to set up a network that would provide timely treatment for both LVO and non-LVO cohorts. Proper balance between the fast IVT treatment for the majority of stroke patients and prompt access to mechanical intervention for MT candidates is crucial. Ideally, all patients should be treated in a closely located institution that provides both treatments in the MS model [29]. However, this solution is practically limited to big agglomerations, while for the patients living in the countryside, transfer times to CSC are often long. Which of the two models (DS vs. MS) provides better outcomes in this scenario is still a matter of debate [33,34]. Most likely, the good results observed in the MS paradigm have a chance to be replicated in the DS model only in the exceptionally well-developed stroke network logistics, which in most cases is unattainable [35,36].

Fast access to a local PSC with prompt IVT is essential for the majority of stroke patients [37]. Yet, for 10%–20% of the most disabling LVO stroke patients, such a model (DS) provides inferior results since delays associated with inter-hospital transportation are considerable [24,32,38].

We believe that stroke networks should undergo a crucial transformation, which might be achieved by creating a middle institution, a thrombectomy-capable stroke centre. It would be based on existing PSC and allow considerable improvement while preserving the best features of both transportation models. TCSC would enjoy inherited PSC’s short time to IVT and bring MT closer to the patients, thus eliminating transfer delays. It is by no means a new concept since the creation of TCSC has been postulated and introduced throughout the world [6,10,11,25,39]. According to recent research, TCSC teams performing above 35 MT per year (15 MT per operator) achieve excellent results, and in the US, they are called ‘high volume’ [6].

Apart from creating a network of TCSC, there is an urgent need for further improvements in stroke services. It is crucial to shorten time intervals in both pre-hospital and in-hospital phases [25]. Neither the Polish Stroke Society in its recommendations on ischaemic stroke treatment nor the European Stroke Organisation (ESO) in the guidelines on mechanical thrombectomy in acute ischaemic stroke specify time limits for acute stroke patient travel time to the hospital [40,41]. The American Heart Association and American Stroke Association in their quality improvement initiative Mission: Lifeline Stroke set the time limit for hospital arrival at 30 min. The National Health Service (NHS) in the UK recommends the optimal transport times to be 30 min and no more than 60 min, and the stroke network should be organised accordingly. In the US, only one-quarter of patients have direct access (within a 15-min drive) to the nearest EVT centre, while approximately 30% of the population live beyond a 1-h travel time [7,25]. Various improvement models have been postulated, and two solutions were investigated: transforming PSC into TCSC or bypassing PSC. The latter option could be contemplated only if the additional time to reach a MT centre is less than 15 min. The result of the modelling process was as follows: when 10% of PSC were transformed into TCSC, fast MT access was achieved for a further 7% of the population. Yet, it was almost doubled for an alternative strategy: bypass transfer to an MT centre [25].

Since it is rather difficult to envisage widespread utilisation of costly mobile CT scanners, the immediate identification of the potential LVO stroke patients should be performed by trained ambulance staff, using validated clinical scales [42]. Even though not supported by the present ESO recommendations, introducing one of them on the national level would certainly simplify training and improve communication between teams. Establishing good lines of communication between stroke centres and ambulances is crucial; the role of neuro tele-consultations is indisputable since it shortens times to treatment [43]. Such a strategy enables the funnelling of patients with large deficits to MT-capable centres, while patients with smaller deficits may be directed to the closest stroke units. Proper patient selection in the field is essential since stroke mimics may constitute up to 25% of patients admitted as acute stroke patients [44].

Another essential step in service improvement would be creating a dense network of MT centres (TCSC and CSC), geared to shorten times to diagnostics and mechanical intervention. Presently, in systems with few MT centres and a dominant DS transportation model, reduction in crucial times proves difficult due to long door-in–door-out (DIDO) times and transport logistics [45,46]. The mothership model provides faster access to invasive treatment but requires the investment in new MT-capable centres and the training of new teams. Since the number of neurointerventional MT operators is highly unsatisfactory, structured training schemes should be introduced, including specialists from different backgrounds with endovascular experience [11,39,47,48,49,50,51].

Since the intervention in acute ischaemic stroke is highly time dependent, it is necessary for healthcare organisers to establish effective MT networks and ensure crucial balance between fast and efficacious treatment and safety. By introducing thrombectomy-capable stroke centres, it is possible to further extend the MT network with the new entity, which has already proven to be safe and effective [12,13]. This will only be possible within a safe environment of teams consisting of neurologists, radiologists, anaesthetists, and interventionists.

We have previously described in detail our experience in creating such a multidisciplinary team, as well as our very early results [15].

Since 2018, when the Mechanical Thrombectomy Pilot Study in Poland was launched, the number of MT centres has been steadily increasing to reach 27 centres per 38 million people in 2022. In 2021, roughly 4% of all Polish acute ischaemic stroke patients were treated by mechanical means [14]. Recognising the mothership transportation paradigm as the closest to ideal, we believe that, in the long term, Poland should strive for a ‘mothership for all’ policy and create a dense network of TCSCs so that patients can receive effective treatment within 1 h of first medical contact. To reduce the time to intervention and provide MT to possibly as many as 12% of all ischaemic stroke patients, Poland needs to triple its network strength and significantly increase the number of MT centres [6,52].

### Limitations of the Study

The main limitation of this study is the relatively small number of patients included and the fact that it was not an all-comer population. We recognize the time difference from symptom onset to randomization (in the HERMES population, up to 12 h; in our group, within 6 h) as a significant limitation. The other shortcoming is the unavailability of the ASPECTS score on the initial CT scan to make a better comparison with the HERMES cohort. Another limitation to consider is the lack of access to the HERMES analysis raw data.

## 5. Conclusions

In this study, we present the results of the first 70 LVO ischaemic stroke patients treated invasively in our small cardiac Cathlab-based TCSC and compare them with the outcomes reported in the intervention arm of the HERMES meta-analysis. Our results, in terms of safety and efficacy, are comparable with those observed in huge neurointerventional centres assessed in HERMES. The encouraging results of this study should inspire further discussions regarding the place of TCSC in future models of acute ischaemic stroke treatment.

## Figures and Tables

**Figure 1 ijerph-20-02232-f001:**
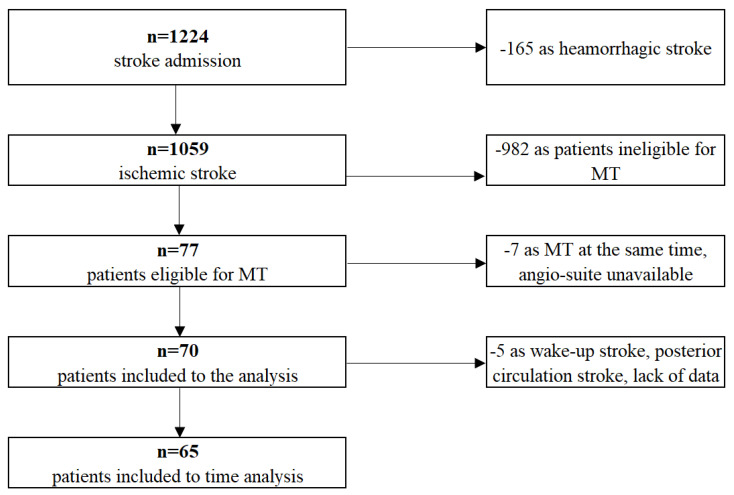
Our group of patients flow chart.

**Table 1 ijerph-20-02232-t001:** Baseline clinical patient characteristics of both groups (mean age, men, women, hypertension, diabetes mellitus, atrial fibrillation, known history of smoking, baseline NIHSS, occlusion location, IV thrombolysis). Data are presented as mean ± SD/N or % (n/N).

		Our cohort	HERMES	*p*
N		70	634	
age (mean ± SD)/N		68.2 ± 12.6/70	67.3 ± 14.7/634	0.6225
women (% (n/N))		42.8% (30/70)	48% (304/634)	0.4942
men (% (n/N))		57.2% (40/70)	52% (330/634)	0.1821
hypertension (% (n/N))		78.6% (55/70)	56% (352/634)	*0.0003*
diabetes (% (n/N))		18.6% (13/70)	13% (82/634)	0.2603
atrial fibrillation (% (n/N))		51.4% (36/70)	33% (209/634)	0.0032
smoking (% (n/N))		25.7% (18/70)	31% (194/634)	0.4788
baseline NIHSS		17 ± 6.2/70	17 ± 4.5/631	0.5915
location	M2	7.1% (5/70)	8% (51/634)	0.9747
	M1	38.6% (27/70)	69% (439/634)	<0.0001
	ICA	62.9% (37/70)	21% (133/634)	<0.0001
	other	1.4% (1/70)	2% (11/634)	0.7653
intravenous thrombolysis (% (n/N))		82.9% (58/70)	83% (526/634)	0.885

**Table 2 ijerph-20-02232-t002:** Process times comparison (three wake-up stroke patients’ data excluded, two patients with missing data excluded). Data are presented as mean ± SD/N.

	Our cohort	HERMES	*p*
onset-to-LVO	104.4 ± 60.2/65	199.4 ± 87.7/632	<0.001
onset-to-IVT	120.1 ± 65.7/55	102.8 ± 43.1/598	0.007
onset-to-reperfusion	272.4 ± 80.7/64	285.7 ± 112.9/634	0.3585

**Table 3 ijerph-20-02232-t003:** Comparison of early efficacy outcomes (mean NIHSS score at 24 h, change in mean NIHSS score from baseline to 24 h, early neurological recovery at 24 h).

	Our cohort	HERMES	*p*
NIHSS at 24 h	10 ± 7.5/67	10.4 ± 8.7/615	0.7175
NIHSS change at 24 h	−6.7 ± 7.4/67	−6.4 ± 8.2/615	0.7742
ENR	46.2% (31/67)	50.2% (309/616)	0.6336

**Table 4 ijerph-20-02232-t004:** Comparison of long-term clinical outcomes (mRS, mRS 0–1, and mRS 0–2 at 90 days) and safety outcomes at 90 days (symptomatic intracranial haemorrhage, parenchymal haematoma type 2, and mortality).

		Our cohort	HERMES	p
mRS	0	14.3% (10/70)	10% (63/634)	0.3544
	1	15.7% (11/70)	16.9% (107/634)	0.9374
	2	25.7% (18/70)	19.1% (121/634)	0.2444
	3	14.3% (10/70)	16.9% (107/634)	0.7013
	4	8.6% (6/70)	15.6% (99/634)	0.1636
	5	4.3% (3/70)	6.2% (40/634)	0.6833
	6	17.1% (12/70)	15.3% (97/634)	0.8177
mRS 0–1		30% (21/70)	26.9% (170/633)	0.6748
mRS 0–2		55.7% (39/70)	46.0% (291/633)	0.1545
sICH		10% (7/70)	4.4% (28/634)	0.0801
PH-2		7.1% (5/70)	5.1% (32/629)	0.6547
mortality		17.1% (12/70)	15.3% (97/633)	0.9568

## Data Availability

The data supporting the report are available from Marek Szołkiewicz (e.mars@wp.pl) on reasonable request.

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
