# Peer review of "Thrombectomy-Capable Stroke Centre—A Key to Acute Stroke Care System Improvement? Retrospective Analysis of Safety and Efficacy of Endovascular Treatment in Cardiac Cathlab"

_ijerph, 2023, doi:10.3390/ijerph20032232_

Round 1
Reviewer 1 Report
The authors report the first results from a new entity, Thrombectomy Capable Stroke Center (TCSC), for the treatment of acute stroke patients with large vessel occlusion. Their results are compared with results from a meta-analysis HERMES comparing five different studies.
The comparison is kept simple using well-known scales such as the NIHSS score at 24h and safety outcomes at 90 days by mRS score. The results are all well presented in a very understandable way. And therefore, I have no comments or corrections to ask for, only two suggestions. Congratulation to the authors for a very nice written manuscript.
1. Could you please refer in the discussion or at the limitations the difference in time to randomization? In the HERMES meta-analysis these were between 6h and 12h, which could have an influence in the outcome.
2. At Table 1. It would be more congruent with the results in column two, to write in column one: “(% (n/N))” instead of “(n/N (%))”.
Author Response
Answers to Reviewer no 1.
Dear Reviewer! Thank you for your time and effort – it is much appreciated.
1. Could you please refer in the discussion or at the limitations the difference in time to randomization? In the HERMES meta-analysis these were between 6h and 12h, which could have an influence in the outcome.
Answer: Thank you very much for this important suggestion, we included reference to it in Limitations.
Limitations of the study
The main limitation of this study is a relatively small number of patients included and the fact that it was not an all-comer population. We recognize time difference from symptom onset to randomization (in the HERMES population up to 12 hours, in our group within 6 hours) as significant limitation. The other shortcoming is the unavailability of the ASPECTS score on initial CT scan to make a better comparison with the HERMES cohort. Another limitation to consider is lack of access to the HERMES analysis raw data.
2. At Table 1. It would be more congruent with the results in column two, to write in column one: “(% (n/N))” instead of “(n/N (%))”.
The changes in Table 1. have been done accordingly.

Reviewer 2 Report
Review of manuscript ijerph- 2134678
Dear authors,
I have read with interest your article entitled: „
Thrombectomy Capable Stroke Center –a key to acute stroke care two system improvement? Safety and efficacy of endovascular three treatment in Cardiac Cathlab“.
It is a crucial topic on the verge of cardiology and neurology. In the US, there are even few interventional neurologists. We definitely need to discuss this topic.
According to your manuscript, there are, however, several issues to be solved in order to meet the high standards of a scientific publication. I do congratulate you on your results and support your interventional enthusiasm, but it has to be written more clearly, and easier to read. I also recommend supporting this manuscript with some images from the intervention showing the recanalization of the cerebral artery.
1) Title: it is not clear what kind of paper you are presenting, is it a original research or review? You have to put it in the title (TCSCT-…: pilot experience nad review of literature? anything)
2) Abstract should be IMRAD structuralized!
3) Data analysis: it should be stated which test you performer and wether the data were normaly distributed, what i really doubt given the number of patients. In case of non-normly distributed data, non-parametric tets should have been utilized.
4) Also the discussion is best , when more structured, i.e. The main findings of our retrospective analysis can be summarized as follows: 1), 2) 3)
Than consider using subheading of disccusion like previous studies/currnt study
5) Limitation of the stufy should be place before the conclusion section
6) The conclusion should shortened and more representative of your own findings
Author Response
Answers to Reviewer no 2.
Dear Reviewer! Thank you for your time and effort – it is much appreciated.
1. Title: it is not clear what kind of paper you are presenting, is it a original research or review? You have to put it in the title (TCSCT-…: pilot experience nad review of literature? anything)
Answer.
Thank you.
To elucidate the character of the article the title has been corrected to:
Thrombectomy Capable Stroke Center – a key to acute stroke care system improvement? Retrospective analysis of safety and efficacy of endovascular treatment in Cardiac Cathlab.
2. Abstract should be IMRAD structuralized!
Answer.
Thanks for this remark.
I reviewed the structure of the abstract according to IMRaD and rephrased accordingly.
3. Data analysis: it should be stated which test you performer and wether the data were normaly distributed, what i really doubt given the number of patients. In case of non-normly distributed data, non-parametric tets should have been utilized.
Thank you for your suggestion, amended as requested.
4. Also the discussion is best , when more structured, i.e. The main findings of our retrospective analysis can be summarized as follows: 1), 2) 3)
Than consider using subheading of discussion like previous studies/currnt study
Answer:
Thank you for this remark.
The main findings of the study were summarized and added at the beginning of this section accordingly.
In this study we compared the early and late safety and efficacy results of the cohort of first 70 LVO stroke patients treated by MT in our centre with the pooled data of 634 patients from the invasive arm of the HERMES study. The main findings of this retrospective analysis may be summarized as follows: 1. the safety outcomes measured at 90days (sICH, PH type 2 and mortality) were comparable in both groups 2. the efficacy outcomes both early (NIHSS at 24h, NIHSS change at 24h and ENR) and late (mRS 0-1 and mRS 0-2 at 90 days) were comparable in both cohorts. In detail: we observed similar ratios of patients with excellent (mRS 0-1) and good (mRS 0-2) clinical outcome at 90 days. Compared to HERMES, a non-significantly higher prevalence of sICH was noted and this may be associated with, present in our cohort, much higher incidence of ICA occlusion [22].
However the details of our analysis and discussion were left unchanged to preserve the clarity of the discussion
5. Limitation of the stufy should be place before the conclusion section
Answer: Thank you for this remark.
The Limitation section was transferred before Conclusions.
6. The conclusion should shortened and more representative of your own findings
Answer.
Thank you.
The Conclusion section was shortened as much as possible.
Conclusions
In this study we present the results of the first 70 LVO ischemic stroke patients treated invasively in our small Cardiac Cathlab based TCSC and compare with the outcomes reported in the intervention arm of the HERMES meta-analysis. Our results, in terms of safety and efficacy, are comparable with those observed in huge neurointerventional centers assessed in HERMES. The encouraging results of this study should inspire the further discussion regarding the place of TCSC in future model of acute ischaemic stroke treatment.

Round 2
Reviewer 2 Report
Dear Authors,
you satisfied the criteria for publication, you did a awesome job. The text of the manuscript is much more readable and clear.
Congratulations